# Land consolidation drives changes in soil bacterial community structure and promotes positive bacterial interactions

Haokun Shi[1,2], Yaoben Lin[3], Siyi Huang[2], Lei Wang[1]*

1 School of Economics and Management, Yiwu Industrial and Commercial College, Yiwu, China,
2 Department of Land Resources Management, School of Public Affairs, Zhejiang University, Hangzhou, China, 3 School of Law and Public Affairs, Nanjing Tech University, Nanjing, China

* wanglei@ywicc.edu.cn

## Abstract

Understanding how land consolidation influences bacterial community structure and interactions is essential for advancing ecological restoration and sustainable land management. However, current knowledge remains insufficient regarding the micro-scale ecological effects of this comprehensive management practice. This study investigated an arable land consolidation project in the Yangtze River Delta plain, where 50 soil samples were collected from consolidated (n = 36) and non-consolidated (n = 14) paddy fields. A comparative analysis was conducted to evaluate the impacts of land consolidation on soil microbial community structure and ecological processes. The findings revealed that land consolidation significantly altered edaphic factors, with moisture content, pH, total nitrogen, and organic matter identified as the dominant drivers of bacterial community structure. Consolidation also reduced the influence of spatial heterogeneity on community composition. Variance partitioning analysis showed edaphic variables (8.84%) and land consolidation (4.34%) significantly contributed to the variations in bacterial community structure. Consolidated fields exhibited a better fit to the neutral community model, suggesting that land consolidation may improve the migration ability of bacterial communities by enhancing soil homogeneity and reducing habitat isolation. The broader niche breadth observed in land consolidation fields further indicated that bacterial communities possess stronger environmental adaptability and community stability, providing a foundation for positive interactions. Interaction analyses revealed higher species coexistence after land consolidation, characterized by stronger positive cohesion and a higher proportion of positive associations in co-occurrence networks. Collectively, these findings suggest that land consolidation could promote positive bacterial interactions, potentially enhancing overall community stability.

**Data availability statement:** All sequence data have been deposited in the NCBI Sequence Read Archive under accession number PRJNA1302308.

**Funding:** This work was supported by the National Natural Science Foundation of China (No. 42301230, Y-B.L.) and the Zhejiang Provincial Philosophy and Social Science Planning (No. 23NDJC373YB, L.W.).

**Competing interests:** The authors have declared that no competing interests exist.

## 1. Introduction

Land consolidation, widely recognized as one of the most effective agricultural management practices for enhancing crop yields, improving ecological conditions, and beautifying rural landscapes [1,2], has been extensively adopted in farmland improvement projects worldwide. The Yangtze River Delta plain, characterized by a long history of agriculture, represents a typical region in which land consolidation has been widely applied to reduce land fragmentation and improve farming practices. In agricultural ecosystems, conventional intensification practices such as tillage and fertilization have long been recognized as major drivers of soil microbial dynamics. Previous studies have largely examined these single management practices, identifying them as major drivers of soil microbial dynamics and discussing their respective advantages and drawbacks [3,4]. In contrast, land consolidation, as a large-scale integrative strategy, simultaneously reshapes farmland landscapes, modifies soil structure, and enhances habitat connectivity, thereby exerting more profound effects on soil ecosystems. Nevertheless, the ecological consequences of land consolidation in arable land remain contentious. While numerous studies have substantiated the benefits of land consolidation in terms of soil fertility and agricultural productivity from a macro-ecological perspective [5], micro-level investigations have highlighted potential short-term risks, including accelerated soil erosion [6] and declines in soil carbon sequestration [7], as consequences of land consolidation. It is well-established that land consolidation measures significantly impact soil microstructure, particularly soil microorganisms [8]. A comprehensive understanding of these mechanisms therefore requires investigation from the microbial ecological perspective [9].

Microorganisms, as the most sensitive component of the soil biological community, are considered one of the most effective indicators of soil quality [10]. For instance, microbial community structure and diversity levels can sensitively reflect external environmental changes [11]. By analyzing microbial indicators, we can elucidate the mechanisms through which land management practices, soil conditions, and other factors influence microbial communities [12]. However, these deterministic factors alone inadequately explain variations in microbial community structure [13]. Microbial ecology emphasizes that assembly mechanisms dominate the evolution of microbial community structure [14] and play a crucial role in predicting and regulating ecosystem functions [15]. Therefore, analyzing microbial community dynamics inherently requires exploring these assembly mechanisms. The relative dominance of stochastic versus deterministic drivers is not fixed [16]. Some studies attribute genetic variation and species diversity to **environmental heterogeneity** (e.g., farming systems, management practices) and spatial distance [17,18], whereas others suggest that microbial assembly in regional paddy soils is primarily regulated by stochastic processes [19]. These insights advance our understanding of how land management practices impact soil microenvironments. Nevertheless, few studies have specifically examined how land consolidation—a critical soil management intervention—affects microbial community assembly processes.

Microorganisms establish complex and stable interactions that are essential for maintaining ecosystem functions [20]. However, soil microbial communities exhibit exceedingly complex dynamics, rendering simplistic assembly patterns or conventional diversity metrics inadequate for characterizing the interconnected relationships. To address this, researchers often employ network-based approaches to analyze microbial community structure and infer species interactions [21]. Empirical evidence demonstrates that certain agricultural practices—such as continuous cropping and monoculture—may diminish network complexity in farmland [22], while land-use transitions from grassland to cropland may paradoxically foster more stabilized microbial interactions to preserve functional redundancy [20]. These findings underscore the need for further research into how land management practices influence microbial networks. Yet, conventional network methods face inherent limitations in resolving the multifaceted nature of microbial interactions, including cooperation, competition, and predation, because these processes are difficult to quantify and model [23]. In contrast, community cohesion metrics serve as a viable alternative to measure the strength and level of complex cross-taxa interactions and community integration, in order to reveal changes in community stability [23].

While previous studies have confirmed the potential of land consolidation to improve soil quality, rice yield and microbial diversity [11,15,20], far less attention has been paid to its effects on community assembly and bacterial interactions. Consequently, there persists a theoretical gap in explaining the mechanistic links between land consolidation and microbial community stability in detail. Thus, this study analyzed bacterial community in consolidated and non-consolidated farmlands using high-throughput sequencing, aiming to: (i) elucidate the effects of land consolidation on bacterial community structure and assembly mechanisms, and (ii) evaluate whether land consolidation facilitates positive interrelationships among bacterial communities. Based on these objectives, we hypothesized that: (i) land consolidation significantly alters soil bacterial community structure, (ii) land consolidation drives shifts in bacterial community assembly mechanisms, and (iii) land consolidation enhances positive bacterial interactions and network cohesion, contributing to greater microbial community stability.

## 2. Materials and methods

### 2.1. Experimental design and soil sampling

This study focused on Jiashan County, located in Jiaxing City, Zhejiang Province, within the Hang-Jia-Hu Plain of the Taihu Lake Basin. The county's geographical coordinates span from 30° 45'36" N to 31° 1' 12" N latitude and 120° 44' 22" E to 121°1'45" E longitude. The region has actively implemented land consolidation projects on suitable arable land plots to improve rural production and living conditions. These land consolidation measures primarily encompass land leveling, field consolidation, water conservancy facility repair, and soil fertilization.

To explore the effect of land consolidation on soil bacterial communities, 50 sampling sites were selected based on official project records provided by the local Land and Resources Bureau in combination with field surveys. Of these sites, 36 were located in consolidation areas that had undergone consolidation for over three years, while the remaining 14 sites were situated in surrounding fields where traditional farming practices had persisted for over a decade without land consolidation (Fig 1). In site selection, additional factors—including soil type, cropping system, environmental conditions, and rice growth status—were considered to ensure comparability between the two groups.

All sampling points, planted with rice and characterized by blue-purple hydromorphic paddy soil, were sampled during three consecutive sunny days. At each point, collect the middle layer (20–40 cm) of soil using a soil auger. A five-point sampling method was employed, consisting of one core from each corner and one from the center of the plot. After removing surface debris, soil samples were homogenized by sieving through a 2-millimeter sieve for removing stones, organic residues, and other debris. The resulting soil samples, classified as Land Consolidation (LC) and Without Land Consolidation (WL), totaling 50, were stored in dry ice within a thermal insulation box and transported to the laboratory at low temperatures. Upon arrival, all soil samples were partitioned into three aliquots: fresh soil (250 g), air-dried soil (250 g), and frozen soil (20 g). Fresh and air-dried soil samples were stored at 4°C for subsequent repeated measurements of edaphic

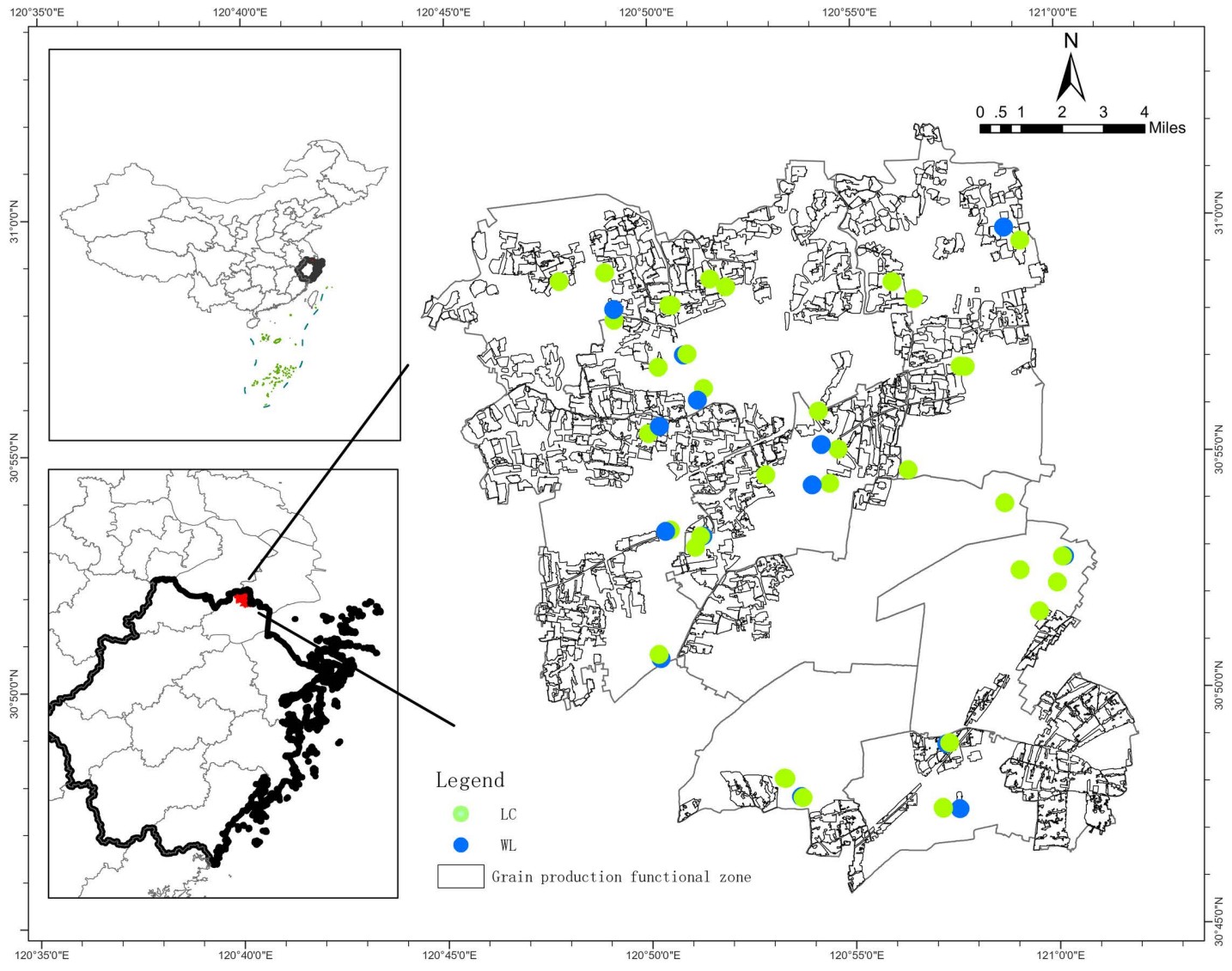

**Fig 1. The study area and sampling sites.**

properties, while frozen samples were stored at −80°C for subsequent DNA extraction [24,25]. To minimize systematic errors, three consecutive edaphic analyses were performed on each soil sample. The edaphic factors of each group are presented in Table 1 (detailed information for all sites is presented in S1 Table). In addition, field blanks were included during sampling to monitor potential contamination. No significant contaminant sequences were detected after quality control, and these controls were excluded from downstream analyses. All soil sampling was conducted with the support of Jiashan County Bureau of Land and Resources.

## 2.2. Soil DNA extraction and sequencing

Microbial data in this study primarily focused on the 16S rRNA prokaryotic community. DNA extraction was conducted using the Fast DNA SPIN Kit (MP Bio-medicals, USA). After extracting DNA of sufficient concentration and quality, the

**Table 1. Edaphic factors of LC and WL samples.**

| Edaphic factor (unit) | LC | WL |
|---|---|---|
| OM (g/kg) | 41.83 ± 14.44a | 30.91 ± 12.23b |
| TN (g/kg) | 2.29 ± 0.67a | 1.53 ± 0.48b |
| AK (mg/kg) | 28.17 ± 13.44a | 32.63 ± 12.27a |
| AP (mg/kg) | 51.40 ± 26.86a | 40.09 ± 27.56a |
| MC (%) | 38.25 ± 7.38a | 33.78 ± 8.57a |
| pH | 6.99 ± 0.57b | 7.86 ± 0.33a |

Significant differences are marked with different lower letters (a and b) at $p < 0.05$.

samples were cryopreserved at −20°C. Subsequent amplification of the 16S rRNA gene's V3-V4 hypervariable region was performed employing the universal primer set 338F/806R (5'-ACTCCTACGGGAGGCAGCA-3'/5'-GGACTACHVGG GTWTCTAAT-3') [26]. For PCR amplification, a 20 µL reaction system was prepared by adding 1 µL of 5 µmol·L$^{-1}$ primer. The PCR program was as follows: 95 °C, 3 min; 25 cycles (95 °C, 30 s, 60 °C, 30 s and 72 °C for 45 s); 72 °C, 10 min; 10 °C, 30 min. Following verification via 2% agarose gel electrophoresis, the amplified DNA fragments were purified employing a gel extraction kit (Omega) to isolate the target DNA fragments. Extraction blanks and PCR blanks were included and sequenced to monitor contamination; no substantial contaminant sequences were detected. Quantification of the purified DNA was performed by Thermo Scientific NanoDrop. Subsequently, the DNA was diluted to 10 ng/µL for further analysis [27,28]. The TruSeq DNA Kit (Illumina) was employed for library construction, and qualified libraries were applied to the Mi Seq system (Mi Seq Reagent Kit V3).

Raw FASTQ files were de-multiplexed with an in-house Perl script, quality-filtered using fastp v0.19.6, and merged by FLASH under the following criteria: (i) reads with an average quality score < 20 over a 50 bp sliding window or length < 50 bp after trimming, as well as reads containing ambiguous bases, were discarded; (ii) only overlapping sequences > 10 bp with ≤ 0.2 mismatch ratio were assembled; (iii) samples were distinguished according to the barcode (exact matching) and primers (≤ 2 nucleotide mismatch). Processing of raw DNA sequencing data was performed utilizing the UPARSE 7.1. Operational taxonomic units (OTUs) with ≥97% sequence similarity were clustered and annotated using the Silva 16S rRNA gene database (Release138, http://www.arb-silva.de). All sequence data have been deposited in the NCBI Sequence Read Archive under accession number PRJNA1302308.

## 2.3. Analysis tools and methods

Non-metric multidimensional scaling (NMDS) was conducted using the "vegan" package in R to visually represent differences in bacterial communities between groups based on Bray–Curtis dissimilarity. To analyze the impact of environmental variables, spatial distance, and land consolidation on bacterial community composition, Mantel tests using the "ggcor" package were conducted to assess Pearson correlations between community compositions and environmental variables. Redundancy analysis (RDA) using the "vegan" package was employed to measure the impact of edaphic factors on bacterial communities. Bray-Curtis distance index was calculated to fit the linear regression relationship between bacterial community similarity and spatial distance, and partial Mantel tests were conducted to control for the effects of measured environmental variables in distance-decay relationship. Principal Co-ordinates of Neighbor Matrices (PCNM) analysis was used to obtain a feature vector matrix characterizing spatial relationships. Based on a forward selection process, using the ordiR2step function, variation partitioning analysis (VPA) was conducted to determine the proportion of bacterial community variation attributable to soil factors, land consolidation, and spatial distance [29]. Significance of PCNM vectors and variance components was assessed using permutation tests (999 permutations). R language was used to implement the Neutral Community Model (NCM) to predict microbial community assembly mechanisms [30], and $\chi^2$ goodness-of-fit

tests were performed to determine whether the observed data significantly departed from NCM. The "spaa" package was employed to calculate community habitat niche breadths, revealing the contribution of dispersal limitation to micro-bial community assembly [31]. Additionally, the "niche.overlap" function was used to calculate niche overlap coefficients between all species pairs [32]. Microbial association networks were inferred using the SPIEC-EASI framework in the pack-age "SpiecEasi", which estimates sparse inverse covariance matrices on centered log-ratio transformed abundance data. A permutation-based null model was implemented to assess the statistical significance of inferred associations. Multiple testing was controlled by applying the Benjamini–Hochberg procedure, and only associations with false discovery rate (FDR) adjusted $p < 0.10$ were retained as significant edges [33]. The resulting networks were visualized in Gephi. Cohe-sion indices were computed in R following the method of Herren and McMahon [23]. Unless otherwise specified, Wilcoxon rank-sum tests were used as the default statistical approach.

## 3. Results

### 3.1. Influencing factors of bacterial community structure and VPA analysis

**3.1.1. The impact of edaphic factors on bacterial communities.** The NMDS results revealed an overlap between the LC and WL samples (Fig 2a), with a stress value of 0.146, indicating an acceptable two-dimensional ordination. Spatially, each group exhibited certain clustering phenomena, indicating that land consolidation significantly influenced bacterial community structure. The results of RDA was illustrated in Fig 2b, revealing that axis RDA1 and RDA2 accounted for 10.72% and 4.33% of the total variance, respectively, with a combined explanatory power of 15.05% of the total variance. Permutation tests (999 permutations) indicated that the structure of bacterial communities was significantly influenced by MC, pH, TN, and OM at the OTU level ($p = 0.001$). Fig 2c presents the correlation analysis between dominant bacterial genera (> 1% relative abundance) and edaphic factors.

**3.1.2. The spatial distance decay relationship of bacterial communities.** It is well-established that bacterial community similarity decreases as geographic distance increases. In this study, a linear regression model was employed to characterize the decay relationship between OTU-level bacterial community similarity and spatial distance (Fig 3). The results revealed a significant negative linear correlation between bacterial community similarity and geographic distance in the total bacterial community (Fig 3a) (Slope = −0.1836, $R^2 = 0.0101$**). A comparison of LC (Fig 3b) and WL (Fig 3c) revealed that the regression slope was shallower in LC (Slope = −0.1401, $R^2 = 0.0064$*) compared to WL (Slope = −0.2513, $R^2 = 0.0082$*), suggesting that structural differences in bacterial communities were more pronounced with increasing distance in WL. However, this relationship became non-significant after controlling for environmental variables in the partial Mantel test, suggesting that the apparent spatial signal was largely attributable to spatially structured environmental gradients.

**3.1.3. The variation partitioning analysis.** Previous research suggest that edaphic factors, land consolidation and spatial distance collectively influence bacterial communities. Consequently, this study utilized VPA to evaluate the relative contributions of LC, edaphic factors, and spatial factors in shaping microbial communities. Employing a forward selection approach, edaphic factors (pH, OM, and AK) and two significant spatial vectors (PCNM17 and PCNM2) were retained for VPA (Fig 4).

The combined factors accounted for 15.39% of the total variation, with the remaining 84.61% unexplained. Permutation variance tests (999 permutations) indicated that both edaphic variables and land consolidation had significant independent effects on bacterial community structure variation. Edaphic variables alone explained 8.84% of the variation (F = 4.4343, $p = 0.001$), while land consolidation alone explained 4.34% (F = 3.3736, $p = 0.001$). Spatial factors, represented by PCNM spatial vectors, did not significantly contribute to the variation in bacterial community structure (F = 1.0296, $p = 0.359$). Moreover, the combined effect of edaphic variables and land consolidation was minimal, explaining only 0.31% of the variation. This suggests that while other environmental variables influenced by land consolidation may contribute to bacterial community structure variation, their overall impact is relatively small.

**Fig 2. The NMDS(a), RDA(b) and correlations(c) between edaphic factors and bacterial community.**

**Fig 3. Geographical distance decay of community similarity of the overall sample(a), LC sample(b) and WL sample(c).** Statistical significance: *p<0.05, **p<0.01, ***p<0.001, the same below.

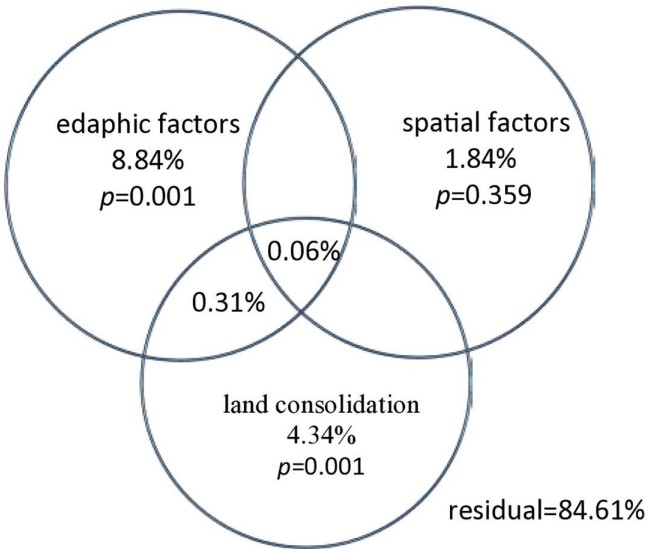

**Fig 4. VPA of the relative contributions of land consolidation, spatial factors, and edaphic factors to community structure.**

### 3.2. The bacterial community assembly network relationships

**3.2.1. Differences in the bacterial community assembly.** VPA results showed that land consolidation, edaphic variables, and spatial factors explained only a small proportion of the variation in bacterial community structure, suggesting that a larger portion of the structural variation may be driven by stochastic processes. Consequently, this study further corroborated the assembly mechanism of bacterial communities.

NCM was employed to predict changes in the occurrence frequency and relative abundance of OTUs. As depicted in Fig 5, the overall bacterial community exhibited a high goodness of fit to the neutral model ($R^2 = 0.742$). Notably, the fit for the LC group ($R^2 = 0.784$) was superior to that of the WL group ($R^2 = 0.628$). The migration rate of bacterial communities in

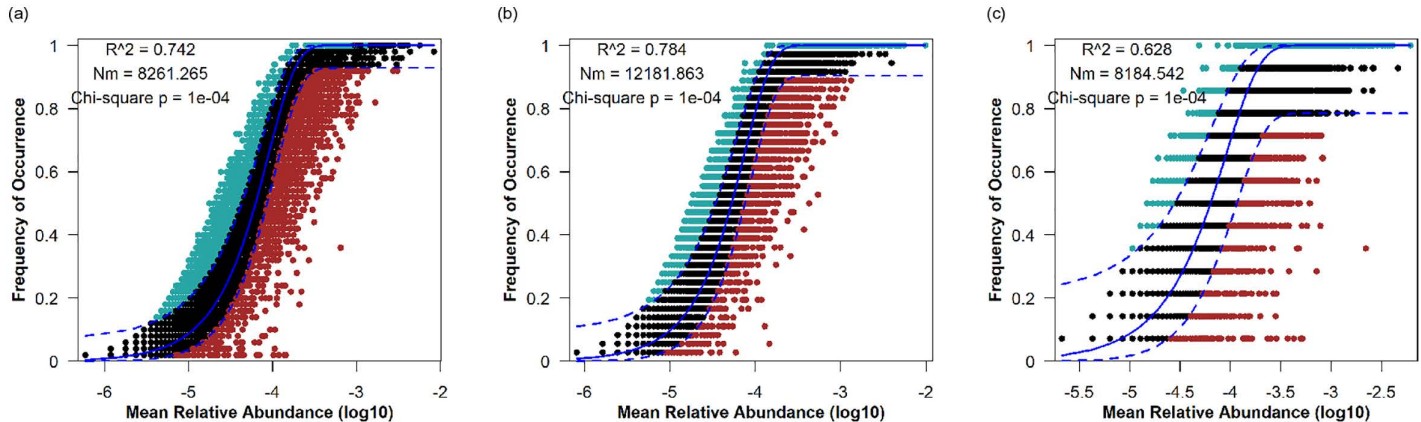

**Fig 5. NCM analysis result of overall sample(a), LC (b) and WL (c).** $R^2$ represents the degree of NCM fit; Nm represents the product of the size of the metacommunity (N) and the migration rate (m).

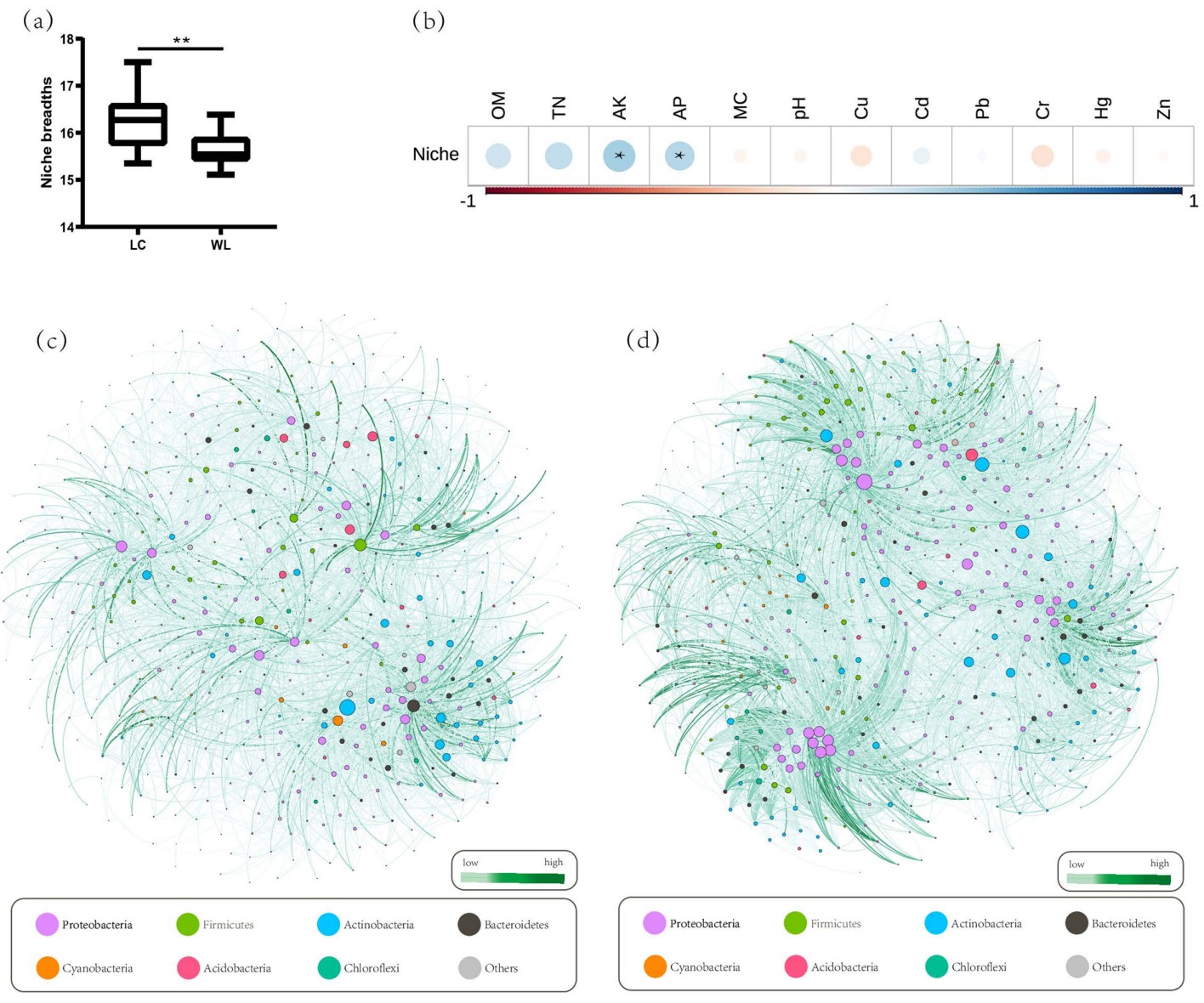

**Fig 6. Niche breadth of various groups(a) and the correlations between niche and edaphic factors(b).** The overlap index network graph of LC(c) and WL(d). Nodes represent the bacterial genera, with size indicating the degree values, and different colors are used to display to which phylum the nodes belong to.

LC (m = 0.358) was higher than that in WL (m = 0.244), indicating enhanced bacterial dispersal in consolidated soils. However, $\chi^2$ goodness-of-fit tests indicated significant deviations between observed and predicted distributions ($p < 0.05$).

Niche breadth prediction helped reveal the contributions of microbial dispersal and coexistence to microbial community assembly [18]. Fig 6a displays a comparison of average niche breadths between various groups, showing that the average niche breadth of LC (16.11 ± 1.00) was significantly greater than that of WL (15.63 ± 0.35, $p < 0.05$). Spearman correlation analysis identified AK and AP as significant predictors of niche breadth (Fig 6b). Further calculation of the niche overlap index between bacterial groups in the soil of LC (0.38 ± 0.29) and WL (0.36 ± 0.31)) at the genus taxonomic level, visualized through network diagrams (Fig 6c and 6d), demonstrated that the WL group exhibited denser ecological niche overlaps and a greater number of connections.

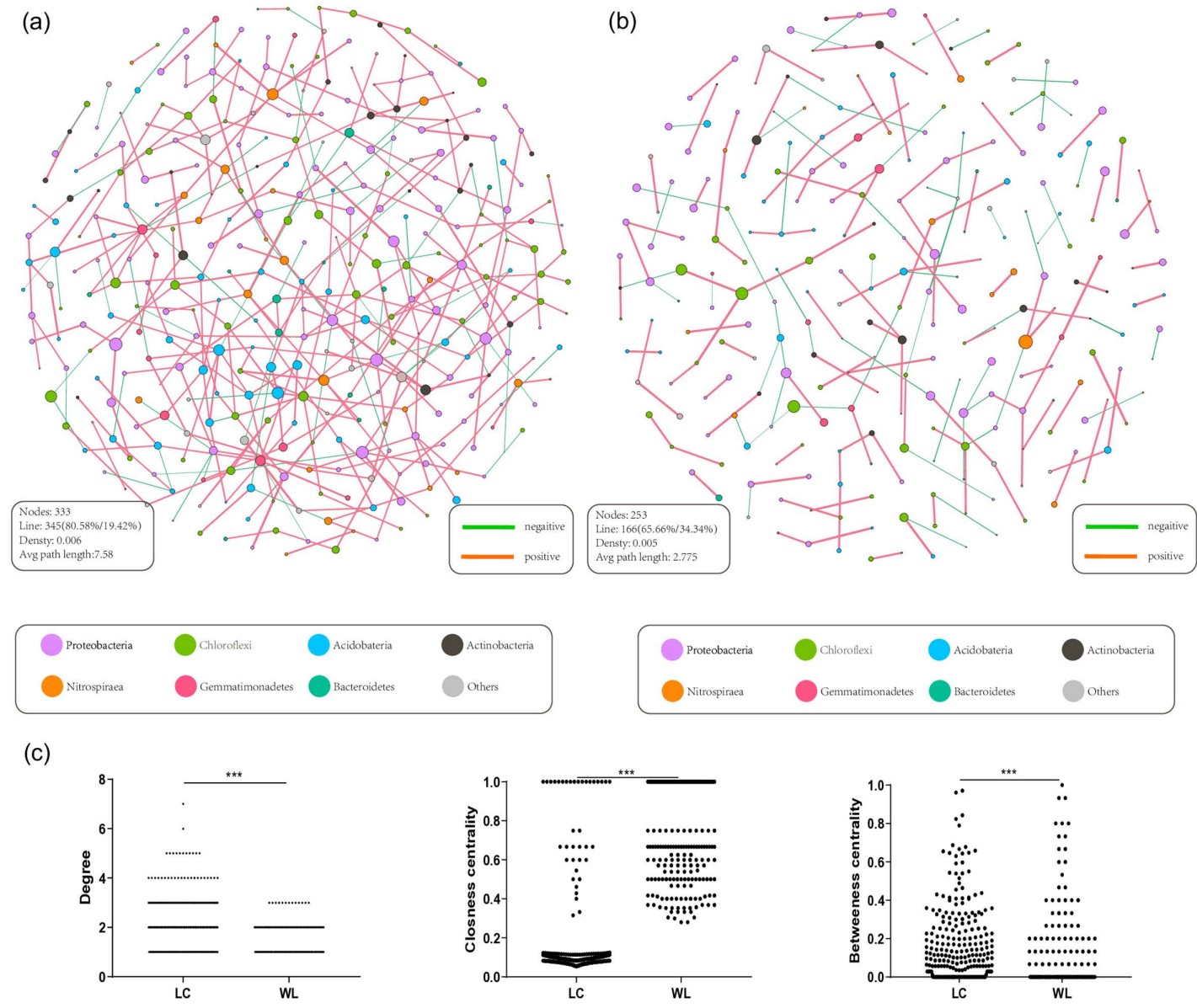

**Fig 7. Bacterial community network based on SPIEC-EASI.** The visualized co-occurrence networks of LC(a) and WL(b). Nodes represent individual OTUs, with colors indicating the phyla to which they belong to. The topological feature comparison at OTU level(c).

**3.2.2. The impact of land consolidation on bacterial community network relationships.** To further assess the impact of land consolidation on the network relationships among bacterial communities, OTU-level co-occurrence networks were constructed, including only OTUs with a relative abundance greater than 0.01% in both LC and WL groups, as visualized in Fig 7a and 7b. The co-occurrence network in LC comprised 333 nodes and 345 edges (FDR-adjusted $p < 0.1$), with 80.58% positive and 19.42% negative correlations. By contrast, the WL network contained 253 nodes and 166 edges, with 65.66% positive and 34.34% negative correlations. Compared to WL, the co-occurrence network in LC exhibited a more complex structure and a greater number of connections, with significantly higher topological features, such as degree ($p < 0.001$) and betweenness centrality ($p < 0.001$). Conversely, closeness

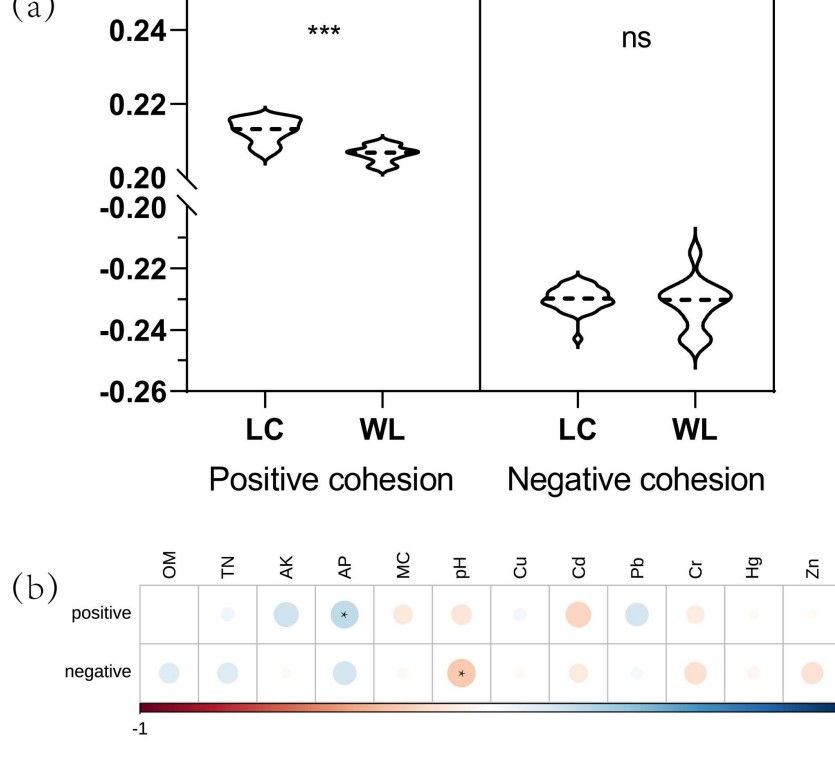

**Fig 8. The community cohesion for various groups(a) and the correlations between community cohesion and edaphic factors(b).**

centrality ($p<0.001$) was higher in WL, as illustrated in Fig 7c. Examination of specific OTU nodes belonging to certain taxa revealed a predominance of negative correlations among dominant species such as *Proteobacteria*, *Chloroflexi*, and *Acidobacteria*.

Cohesion analysis further highlighted differences between LC and WL (Fig 8a). Positive cohesion was significantly higher in LC ($0.2127\pm0.0031$) than in WL ($0.2067\pm0.0026$; $p<0.001$), whereas negative cohesion showed no significant difference. Spearman correlation analysis (Fig 8b) further revealed that positive cohesion was primarily associated with AP ($p<0.05$), whereas negative cohesion showed associations with pH ($p<0.05$).

## 4. Discussion

### 4.1. Land consolidation, spatial factors, and edaphic factors jointly drive changes in bacterial community structure

NMDS analysis confirmed the significant impact of land consolidation on bacterial community structure ($p=0.001$), but other studies have also indicated that environmental heterogeneity and spatial distance are widely recognized as the main factors contributing to genetic variation and population diversity. RDA further supported that TN and OM exerted significant effects ($p=0.001$) on bacterial communities. Pearson correlation tests also showed that few bacterial taxa were significantly correlated with AP or AK, whereas TN and OM were positively correlated with multiple groups, such as *Nitrospira* (involved in nitrogen cycling) and *Thiobacillus* (involved in sulfur and carbon cycling) (Fig 2c) [34,35]. Based on the synergistic mechanism of nutritional interactions, it was inferred that, in the study area, TN and OM were more critical edaphic factors limiting crop growth compared to other nutrients [36]. Therefore, optimizing fertilizer ratios to balance soil nutrients may be necessary for sustainable cultivation. RDA also indicated that MC and pH were primary determinants

shaping bacterial community structure, and maintaining soil moisture is conducive to ensuring microbial activity and community diversity levels [19]. pH is considered an important indicator for analyzing and predicting soil microbial community structure, consistent with previous studies on paddy field areas [37,25]. This study further substantiated the significant impact of pH on bacterial community structure, although its impact on microorganisms remains debated [38], with further work needed to determine the optimal pH range across regions.

An important question in biogeography is how spatial distance influences the assembly of bacterial communities. Although previous studies have consistently demonstrated an inverse relationship between spatial distance and bacterial community similarity [39], our findings showed that this pattern was weak in LC soils within the small spatial scale and became non-significant after controlling for environmental variables. This suggests that the apparent spatial effect was largely confounded by environmental gradients that are spatially structured. Land consolidation projects, through measures such as field consolidation and land leveling, likely reduce habitat isolation and promote soil homogenization [40]. In addition, uniform farming practices and intensive chemical inputs may further stabilize soil niches [18], thereby diminishing the influence of geographic distance on community turnover.

### 4.2. Land consolidation can affect bacterial community assembly

The exploration of microbial community assembly mechanisms has been a longstanding focus in microbial ecology. Niche theory and neutral theory represent two complementary mechanisms of microbial community assembly. Niche theory posits that biological communities are regulated by deterministic processes involving abiotic factors (such as edaphic variables) and biotic factors [41]. In contrast, neutral theory suggests that random processes, including stochastic birth, death, and migration events, shape the structure of microbial communities [31]. In this study, VPA results indicated a substantial proportion of unexplained bacterial community variation, suggesting the potential significance of stochastic processes in community assembly [42]. Consequently, NCM and habitat niche breadth analysis were employed to infer the assembly and migration processes of bacterial communities.

NCM accounted for 74.2% of the overall variation in the bacterial community, indicating the dominance of stochastic processes in the assembly of these communities. The migration rate and fitness of the bacterial communities in LC ($R^2 = 0.784$, $m = 0.358$) were both higher than those in WL ($R^2 = 0.328$, $m = 0.244$), suggesting greater dispersal capabilities for bacterial communities after consolidation. A plausible explanation is that, as bacteria are prone to migrate and diffuse with water flow [30], the LC areas, equipped with improved water conservancy facilities, are more likely to experience regular wet–dry cycles and extensive irrigation. Longer water retention also substantially weakens microbial dispersal limitation [18]. In addition, land leveling reduces habitat isolation, further alleviating dispersal constraints [40]. Previous studies have reported that land consolidation projects can enhance the connectivity and homogeneity of arable soils by reshaping soil structure and creating more uniform physicochemical conditions favorable to stochastic processes [43]. Nevertheless, it should be noted that the $\chi^2$ goodness-of-fit tests revealed significant deviations between observed and predicted distributions ($p < 0.05$), indicating that although the neutral model captures broad patterns, it cannot fully account for community dynamics. For example, homogenized soil physicochemical conditions may also strengthen deterministic processes [44].Niche breadth reflects environmental adaptability and dispersal ability, with higher values indicating greater survival capacity and broader ecological ranges. Different farming practices create heterogeneous soil environments, promoting significant changes in microbial community structure and phylogeny [45]. The niche breadth in LC was significantly greater than in WL ($p < 0.05$), suggesting that bacterial communities in consolidated soils face fewer dispersal limitations and exhibit stronger environmental adaptability. Indeed, stable farming practices in LC may also select for bacterial taxa with high fitness under uniform conditions, thereby diminishing the role of environmental filtering [46]. By contrast, communities with lower niche breadths in WL areas are more strongly shaped by environmental selection pressures and external disturbances, reducing overall stability.

## 4.3. Land consolidation can promote cooperation between bacterial communities

The analysis of niche overlap indices revealed significant differences between LC and WL. Examination of species-specific topological features in the co-occurrence networks showed greater niche overlap in WL, suggesting fiercer competition among species for space and resources in this area [47]. By contrast, bacterial taxa in LC exhibited stronger metabolic capabilities, physiological tolerance, rapid growth, and dispersal migration abilities, potentially occupying broader ecological niches and facilitating the coexistence of more species within the meta-community. Interestingly, taxa with high niche overlap indices did not always have high node degrees, a phenomenon particularly evident in the co-occurrence network of WL. This may imply that such taxa face more intense competition from a limited number of groups for environmental resources. This suggests that land consolidation may provide more resources and space, supporting the metabolic needs of diverse bacterial groups. Correlation analysis of niche breadth with corresponding edaphic properties (Table 2) indicated a significant positive correlation between niche breadth and AK and AP. Previous studies have shown that phosphorus availability strongly influences microbial niches, shaping different assembly patterns in abundant and rare bacterial communities in forest soils [48]. Similarly, Similarly, variation in potassium utilization rates can promote microbial coexistence and diversification of ecological niches [49], potentially influencing the metabolism and adaptability of microbial groups.

Topological feature indices can effectively quantify the impact of land consolidation on microbial networks, particularly in terms of micro-ecosystem stability [45]. In the LC sub-network, bacterial nodes exhibited higher degrees and stronger betweenness centrality, indicating that consolidation promoted more extensive microbial interactions and the emergence of key "bridge" taxa that facilitate communication across different modules. Such increased structural complexity enhances the robustness of microbial networks by enabling efficient information transfer and buffering against environmental disturbances [50]. In contrast, the WL network showed higher average closeness centrality, suggesting shorter overall paths and more locally compact connections. However, this apparent tightness reflects a more centralized and less diverse structure, which may be vulnerable to disruptions if a few nodes or pathways are lost.. Overall, this study found that the sub-network of LC exhibited higher stability compared to WL. Topological parameters revealed higher complexity and connectivity in LC, suggesting that land consolidation enhances the resilience of soils to environmental disturbances and improves their capacity to withstand ecological risks [51]. Positive cohesion reflects cooperative interactions among taxa driven by mutualistic symbiosis or niche complementarity, whereas negative cohesion signifies antagonistic interactions arising from resource competition or allelopathic effects [52]. Cohesion metrics analysis revealed significantly higher

Table 2. Pearson's correlation coefficients (r) between α-diversity metrics of bacterial community and edaphic factors of samples.

| Edaphic factor | Sobs | Shannon | Ace | Pd | Coverage |
|---|---|---|---|---|---|
| OM | 0.142 | 0.161 | 0.169 | 0.097 | 0.157 |
| TN | 0.130 | 0.160 | 0.170 | 0.056 | 0.124 |
| AP | −0.084 | −0.028 | −0.065 | −0.114 | −0.04 |
| AK | **−0.302*** | **−0.300*** | **−0.307*** | −0.192 | −0.067 |
| MC | **0.328*** | **0.340*** | **0.349**** | **0.286*** | 0.153 |
| pH | −0.004 | 0.040 | −0.044 | 0.070 | 0.073 |
| Cu | 0.111 | 0.136 | 0.061 | 0.156 | 0.077 |
| Cd | 0.167 | 0.178 | 0.111 | 0.159 | 0.124 |
| Pb | 0.106 | 0.076 | 0.077 | 0.084 | 0.058 |
| Cr | 0.250 | 0.225 | 0.21 | 0.239 | 0.081 |
| Hg | 0.089 | 0.048 | 0.034 | 0.092 | 0.027 |
| Zn | 0.182 | 0.170 | 0.140 | 0.188 | 0.103 |

Data in bold indicate significant correlations, *$p < 0.05$, **$p < 0.01$.

positive cohesion in LC compared to WL ($p<0.05$), with no significant difference in negative cohesion. This divergence likely stems from improved soil nutritional conditions after consolidation: the enriched nutrient profile in consolidated soils may sustain functionally redundant bacterial taxa through metabolic specialization, including niche sharing and coexistence among microbial groups involved in nutrient cycling and decomposition [53], thereby generating consistent positive cohesive forces. Notably, a significant positive correlation emerged between positive cohesion and AP content ($p<0.05$), aligning with ecological theories positing that resource availability promotes coexistence of functionally redundant taxa [54]. Conversely, pH exhibited a significant negative correlation with negative cohesion ($p<0.05$). The inverse relationship may originate from acidic environmental filtering of specific functional guilds. As demonstrated by Kielak, *Acidobacteria* act as K-strategists with competitive advantages in oligotrophic environments [55], potentially intensifying negative interactions through nutrient competition [54]. Such pH-mediated resource competition could account for the observed inverse association between negative cohesion and pH.

### 4.4. Limitations and future perspectives

While this study provides insights into the effects of land consolidation on soil bacterial communities, several limitations should be considered. First, it must be acknowledged that the sampling design was unbalanced, although subsampling analyses (999 iterations) were conducted to confirm robustness (S1 Fig, S3 Table), such imbalance may still reduce statistical power and increase the potential bias. Second, our dataset represents a single temporal snapshot and does not capture seasonal or interannual dynamics. Third, VPA indicated that 84.61% of community variation remained unexplained, likely reflecting unmeasured soil factors such as organic pollutants or dissolved organic carbon. Finally, this study examined only bacterial communities; future research should extend to other microbial groups (fungi, archaea, viruses) to achieve a more comprehensive understanding of soil microbial ecology under land consolidation. Addressing these limitations through expanded sampling designs, multi-omics approaches, and long-term monitoring will help advance a more mechanistic understanding of how land consolidation shapes soil microbial communities.

### 5. Conclusions

This study provides new insights into the ecological impacts of land consolidation on bacterial community structure, assembly mechanisms, and network interactions in paddy soils of the Yangtze River Delta plain. Land consolidation improves soil physicochemical properties and reduces the influence of spatial heterogeneity on bacterial community structure through unified land management measures. Variation partitioning analysis demonstrated that both land consolidation and edaphic factors were major drivers of bacterial communities, yet the substantial unexplained variation underscored the importance of stochastic processes. Neutral model and niche analyses further revealed enhanced dispersal capacity, broader ecological niches, and greater environmental adaptability in consolidated soils. In addition, co-occurrence network and cohesion analyses showed that land consolidation strengthened positive microbial interactions, reinforced symbiotic relationships, and improved community stability. Taken together, these results offer a mechanistic understanding of how land consolidation regulates soil microbial ecology. They highlight its dual role in reshaping soil environments and fostering sustainable agricultural development, thereby providing a theoretical foundation for integrating land management with ecological restoration strategies.

### Supporting information

**S1 Fig. Distributions of key ecological metrics across 999 balanced subsampling iterations.**
(TIF)

**S2 Fig. Conceptual framework showing how land consolidation influences soil factors.**
(TIF)

**S3 Fig. Bacterial community network relationship based on spearman correction.**
(PDF)

**S1 Table. Detailed soil physicochemical properties of all sampling sites.**
(CSV)

**S2 Table. Comparison of soil heavy metal concentrations between LC and WL.**
(XLSX)

**S3 Table. Summary of key ecological metrics across 999 balanced subsampling iterations.**
(CSV)

**S4 Table. IQR-based thresholds of soil physicochemical properties for sensitivity analysis.**
(XLSX)

**S5 Table. Sensitivity analysis comparing soil edaphic properties and bacterial community metrics between LC and WL.**
(XLSX)

## Acknowledgments

We extend our gratitude to Shandong Wudi Golden Land Environmental Testing Co., Ltd. for their assistance in sampling.

## Author contributions

**Data curation:** Haokun Shi.

**Formal analysis:** Haokun Shi, Siyi Huang.

**Funding acquisition:** Yaoben Lin, Lei Wang.

**Methodology:** Yaoben Lin.

**Project administration:** Lei Wang.

**Resources:** Lei Wang.

**Software:** Haokun Shi.

**Validation:** Yaoben Lin, Siyi Huang.

**Writing – original draft:** Haokun Shi.

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
