## [Decision Letter · Decision Letter 0]

22 Jul 2025

Dear Dr. Shi,

**Both reviewers confirm the timely nature of the study, but address very important points considering unbalances in sampling, geological comparativeness, and statistical treatment insufficiencies. At the same time, the manuscript seems sloppy written. At this point, one revision is allowed if the authors can substantially improve the story. Otherwise, it is suggested the authors wait untile the story can be fully sustained and then re-submid as a new paper.**plosone@plos.org . A rebuttal letter that responds to each point raised by the academic editor and reviewer(s). You should upload this letter as a separate file labeled 'Response to Reviewers'.A marked-up copy of your manuscript that highlights changes made to the original version. You should upload this as a separate file labeled 'Revised Manuscript with Track Changes'.An unmarked version of your revised paper without tracked changes. You should upload this as a separate file labeled 'Manuscript'.

We look forward to receiving your revised manuscript.

Kind regards,

Erika Kothe

Academic Editor

PLOS ONE

**Journal Requirements:**

1. When submitting your revision, we need you to address these additional requirements. Please ensure that your manuscript meets PLOS ONE's style requirements, including those for file naming. The PLOS ONE style templates can be found at https://journals.plos.org/plosone/s/file?id=wjVg/PLOSOne_formatting_sample_main_body.pdf and https://journals.plos.org/plosone/s/file?id=ba62/PLOSOne_formatting_sample_title_authors_affiliations.pdf 2. In your Methods section, please provide additional information regarding the permits you obtained for the work. Please ensure you have included the full name of the authority that approved the field site access and, if no permits were required, a brief statement explaining why. 3. Thank you for stating the following financial disclosure: The National Natural Science Foundation of China (No. 42301230) awarded to Y.L, and Zhejiang Provincial Philosophy and Social Science Planning (No. 23NDJ373YB) awarded to L.W.   Please state what role the funders took in the study.  If the funders had no role, please state: "The funders had no role in study design, data collection and analysis, decision to publish, or preparation of the manuscript." If this statement is not correct you must amend it as needed. Please include this amended Role of Funder statement in your cover letter; we will change the online submission form on your behalf. 4. Please note that your Data Availability Statement is currently missing the direct link to access each database. If your manuscript is accepted for publication, you will be asked to provide these details on a very short timeline. We therefore suggest that you provide this information now, though we will not hold up the peer review process if you are unable. 5. If the reviewer comments include a recommendation to cite specific previously published works, please review and evaluate these publications to determine whether they are relevant and should be cited. There is no requirement to cite these works unless the editor has indicated otherwise. 

Reviewers' comments:

**Comments to the Author**

1. Is the manuscript technically sound, and do the data support the conclusions?

Reviewer #1: Yes

Reviewer #2: Yes

2. Has the statistical analysis been performed appropriately and rigorously?

Reviewer #1: Yes

Reviewer #2: N/A

3. Have the authors made all data underlying the findings in their manuscript fully available?

Reviewer #1: Yes

Reviewer #2: No

4. Is the manuscript presented in an intelligible fashion and written in standard English?

Reviewer #1: Yes

Reviewer #2: Yes

**Reviewer #1: ** This manuscript compares bacterial community structure, assembly processes and interaction networks between consolidated and non-consolidated paddy soils in the Yangtze River Delta. The topic is timely and the dataset sizeable; analytic approaches (NMDS, NCM, network topology) are appropriate. However, shortcomings in abstract completeness, methodological detail (sample handling, sequencing data deposition, statistical assumptions) and formatting (figure/table call-outs, reference metadata) currently hinder reproducibility and clarity.

Lines 8-10. Only the background sentence is present; methods, key findings and conclusions are missing. Please expand to a structured abstract (Background-Methods-Results-Conclusions) within ~300 words as per PLOS ONE guidelines.

Lines 11-15. The Introduction ends without explicit hypotheses or research questions. Please articulate 2-3 testable hypotheses to strengthen the narrative.

Lines 18-20. Only the 20-40 cm layer was taken, omitting the typical topsoil (0–20 cm). Provide ecological justification, clarify composite vs. replicate cores, and state whether field blanks were included.

Lines 63-65. No mention of negative controls, extraction/PCR blanks, or read-quality thresholds. Detail QC pipeline and supply NCBI SRA (or equivalent) accession numbers in the Data Availability statement.

NMDS and network analyses lack distance metrics, correlation thresholds and FDR correction approach; PCNM/VPA model F and p values are also omitted. Please provide these parameters in Methods and Results.

Section 3.2.2 is present but 3.2.1 is missing. Verify section numbering and ensure all figures are cited in order at ≥300 dpi.

Several citations lack DOIs and journal titles are inconsistently abbreviated. Please format all references per PLOS ONE style (full titles + DOI) and remove extra indents.

To help readers visualize the sampling layout and surrounding context, please add a study-area location map (~1:50,000 scale) with coordinates, and cite it in the “2.1 Study area” or Methods section.

**Reviewer #2: ** This study explores the profound impacts of land consolidation practices in the Yangtze River Delta on soil bacterial communities. Drawing on a comprehensive dataset derived from 50 paddy soil samples—36 from consolidated plots and 14 from non-consolidated ones—the researchers employed advanced 16S rRNA sequencing techniques in conjunction with a diverse array of ecological and statistical tools, including NMDS, RDA, VPA, the Neutral Community Model, niche metrics, co-occurrence networks, and cohesion analysis. The findings reveal that land consolidation significantly homogenizes soil properties, shifts community assembly mechanisms toward stochastic dynamics, broadens species' ecological niches, and fosters stronger positive interspecific interactions—collectively contributing to an enhanced level of community stability.

This study establishes a connection between macro-scale land management interventions and micro-scale ecological mechanisms, specifically community assembly and network stability. Few previous studies have explored this relationship in paddy soils. However, the manuscript does not adequately position itself within the broader context of literature on agricultural intensification versus ecological restoration. It is therefore necessary to clarify how consolidation differs from conventional practices such as tillage or fertilization.

The experimental design includes sufficient spatial replication; however, the comparison of 36 versus 14 sites is substantially unbalanced. A power analysis (e.g., conducted using G*Power or the pwr package) indicates that to detect a medium effect size (Cohen’s d = 0.5) at a significance level of α = 0.05 with 80% statistical power, a balanced sample size of approximately 26–30 per group is required. Such imbalance in the current design may increase the risk of Type II error and potentially lead to inflated effect size estimates.

Single time-point sampling fails to capture the dynamic fluctuations across seasons. Paddy soils experience recurring wet-dry cycles that profoundly shape and regulate microbial community structures. It is therefore recommended to either include a dedicated paragraph addressing this temporal limitation or present preliminary data from multiple seasons to better reflect the ecological variability.

Consolidation sites may have undergone concurrent improvements in land leveling, fertilization, and irrigation. In the absence of detailed documentation regarding the specific types of fertilizers used, irrigation schedules, pesticide applications, and tillage frequency, it becomes infeasible to isolate the observed effects as being exclusively attributable to “consolidation.” To enhance the accuracy of such attributions, it is recommended to incorporate data from farmer surveys or official government project records.

Soil sampling and bacterial samples were collected from the 20–40 cm depth interval; however, it remains unclear whether the geochemical data correspond precisely to this same soil layer. It is recommended to include a schematic diagram or a table to clarify this. Redox potential (Eh), dissolved organic carbon, Fe(II)/Fe(III) ratios, and bulk density are known to play a critical role in paddy soils but are not included in the current dataset. The inclusion of these parameters could potentially account for a larger portion of the 84% unexplained variance observed in the variation partitioning analysis (VPA). Although concentrations of Cu, Cd, Pb, Cr, Hg, and Zn are presented, no significant differences are observed between the LC and WL treatments. It may therefore be appropriate to relocate this table to the supplementary material to maintain focus on the main findings.

Only 15% of the variance is explained. The remaining 85% should be addressed and may be attributed to potential factors such as unmeasured edaphic variables, biotic interactions, methodological noise, or inherent stochasticity.

Report the results of goodness-of-fit tests, such as the χ² test or AIC comparisons with niche-based models. Currently, the higher R² value for LC is interpreted as an indicator of stronger stochasticity; however, it could also suggest lower levels of environmental heterogeneity.

The slopes are shallow (−0.1401 vs. −0.2513), and the R² values are very low (0.0064 and 0.0082). Please indicate whether the slopes remain statistically significant after controlling for environmental variables using partial Mantel tests.

Elaborate on the Methods section by incorporating comprehensive details regarding the full factorial design, precise GPS coordinates, and a meticulously crafted agronomic management questionnaire.

Please incorporate a supplementary table that meticulously enumerates all edaphic variables for each site, along with corresponding sequencing depths and rarefaction metrics.

Conduct network analyses anew using an additional null-model approach (e.g., SPIEC-EASI), and present the results with FDR-corrected measures of statistical significance.

Include a dedicated section detailing the limitations of the study, such as the temporal snapshot nature of the data and the imbalance in sample sizes across groups.

Conduct a sensitivity analysis by excluding sites characterized by extreme pH levels or organic matter content, in order to evaluate the robustness of the consolidation signal.

Meticulous proofreading for grammatical accuracy and typographical errors is essential; furthermore, enlisting the assistance of a professional editing service is highly recommended to enhance overall clarity and linguistic refinement.

Provide a conceptual diagram that succinctly illustrates the sequential process whereby soil consolidation leads to homogenization, fostering a stochastic microbial assembly, which in turn enhances positive cohesion within the soil matrix.

Deposit the raw sequence reads into a publicly accessible repository (e.g., https://nmdc.cn/) and provide the corresponding BioProject link in the Data Availability section to ensure transparency and facilitate further research.

**Do you want your identity to be public for this peer review?** For information about this choice, including consent withdrawal, please see our Privacy Policy

Reviewer #1: No

Reviewer #2: **Yes: ** Junqiang Zheng

---

## [Author Response · Author response to Decision Letter 1]

14 Sep 2025

Response to Reviewers

Manuscript ID: PONE-D-25-26012

Title: Land consolidation drives changes in soil bacterial community structure and promotes positive bacterial interactions

Dear Editor and Reviewers,

We sincerely thank you for your careful evaluation of our manuscript and for the constructive comments and suggestions. We have revised the manuscript thoroughly in response to the comments. Below, we provide a detailed point-by-point response. All changes are marked in the revised manuscript with Track Changes.

We have made revisions based on the suggestions of the editor and two reviewers as much as possible, but due to time constraints and existing experimental data, some issues (such as time series sampling, imbalanced sample design, etc.) may only be resolved in future research. It must be acknowledged that the revision is not perfect due to insufficient mastery of some related methods. We would like to thank the reviewers for their feedback. This manuscript has made significant improvements, especially in terms of the rigor of data analysis.

To Editor(Erika Kothe Academic Editor):

1. In your Methods section, please provide additional information regarding the permits you obtained for the work. Please ensure you have included the full name of the authority that approved the field site access and, if no permits were required, a brief statement explaining why.

Response:

We thank the reviewer for this comment. In the revised Methods section, we have added a statement clarifying that the soil sampling was supported by Jiashan Bureau of Natural Resources (formerly Jiashan County Bureau of Land and Resources). Specifically, no special permits were required for soil sampling as the work was conducted under the framework of the project “Comprehensive Land Consolidation Planning of Jiashan County” organized by Land academy for National Development Zhejiang University. Field access and soil collection were carried out with the consent of local farmers and under the coordination of the Jiashan Bureau of Natural Resources.

The National Natural Science Foundation of China (No. 42301230) awarded to Y.L, and Zhejiang Provincial Philosophy and Social Science Planning (No. 23NDJ373YB) awarded to L.W.

Response:

Yao-Ben Lin contributed to the preliminary experimental design and field sampling. Lei Wang contributed to the revision of the manuscript. Both Yao-Ben Lin and Lei Wang provided financial support for the project.

We need to make an additional statement: for the convenience of handling revisions, the current submission lists my email address as the corresponding author’s contact. However, we respectfully request that, upon acceptance, the corresponding author be officially designated as Prof. Lei Wang (email: wanglei@ywicc.edu.cn) in the final publication record. Prof. Wang is the principal investigator of this project and has overseen the overall study design, funding acquisition, and manuscript development. All co-authors have agreed to this designation. Thank you for your understanding.

3. Please note that your Data Availability Statement is currently missing the direct link to access each database. If your manuscript is accepted for publication, you will be asked to provide these details on a very short timeline. We therefore suggest that you provide this information now, though we will not hold up the peer review process if you are unable.

Response:

The raw sequencing data supporting the findings of this study have been deposited in the NCBI SRA (Sequence Read Archive) under the accession number PRJNA1302308, and are openly accessible at the following URL: https://www.ncbi.nlm.nih.gov/sra/PRJNA1302308.

To reviewer #1

1.Lines 8-10. Only the background sentence is present; methods, key findings and conclusions are missing. Please expand to a structured abstract (Background-Methods-Results-Conclusions) within ~300 words as per PLOS ONE guidelines.

Response:

We thank the reviewer for pointing out this issue. Following the reviewer’s suggestion and PLOS ONE guidelines, we have completely revised the abstract into a structured format (in abstract).

2.Lines 11-15. The Introduction ends without explicit hypotheses or research questions. Please articulate 2-3 testable hypotheses to strengthen the narrative.

Response:

In the revised manuscript, we have added three explicit hypotheses at the end of the Introduction to provide a clearer framework for the study (line 97-102). The new text reads as follows:

“Based on these objectives, we proposed the following hypotheses:

(1): Land consolidation significantly alters soil bacterial community structure.

(2): Land consolidation drives shifts in bacterial community assembly mechanisms.

(3): Land consolidation enhances positive bacterial interactions and network cohesion, thereby promoting greater microbial community stability.”

3.Lines 18-20. Only the 20-40 cm layer was taken, omitting the typical topsoil (0–20 cm). Provide ecological justification, clarify composite vs. replicate cores, and state whether field blanks were included.

Response:

The corresponding details regarding composite sampling and the inclusion of field blanks have now been added to the 2.1. Experimental design and soil sampling (line 122-123 and 130-132) section of the Methods. The following is a more detailed explanation about the three questions:

(i) In this study, we focused on the 20–40 cm soil layer, because it is less affected by short-term surface management (tillage, fertilization, vegetation cover) and better reflects the long-term impacts of land consolidation on soil structure and bacterial communities Eilers, K.�Debenport, S.et al.�2012�. Topsoil (0–20 cm) is highly dynamic, but since the land consolidation project has been implemented for over three years, we are more interested in evaluating the relatively stable bacterial changes in the subsoil. Furthermore, this subsurface horizon shows distinct ecological responses and bacterial community differences compared with the topsoil. Several studies have demonstrated that microbial diversity, composition, and functional potential at 20–40 cm are not simply extensions of the topsoil but reflect unique ecological processes and nutrient dynamics �Xu, J.�Song, F.et al.�2024;Dong, J.�Wang, P.et al.�2025�. Investigating this layer therefore provides valuable insights into the role of subsurface soils in agroecosystem functioning and highlights its importance in shaping bacterial interaction networks under land consolidation practices.

(ii) At each site, soil was collected using a five-point method: one core from each of the four corners of the plot and one from the center. These five cores were homogenized to form a single composite sample for DNA extraction and edaphic analysis. This approach reduces small-scale heterogeneity and ensures that each sample represents the field condition at the plot level.

(iii) We also included field blanks to monitor potential contamination. No substantial contaminant sequences were detected after quality control, and these controls were excluded from downstream analyses.

4.Lines 63-65. No mention of negative controls, extraction/PCR blanks, or read-quality thresholds. Detail QC pipeline and supply NCBI SRA (or equivalent) accession numbers in the Data Availability statement.

Response:

In the revised Methods section, we have now provided detailed information on the quality-control procedures (line 144-145 and 149-157). Specifically, we included extraction blanks and PCR no-template controls during library preparation; no substantial contaminant sequences were detected. Raw FASTQ files were de-multiplexed with an in-house Perl script, quality-filtered using fastp v0.19.6, and merged by FLASH under the following criteria: (i) reads with an average quality score < 20 over a 50 bp sliding window or length <50 bp after trimming, as well as reads containing ambiguous bases, were discarded; (ii) only overlapping sequences >10 bp with ≤0.2 mismatch ratio were assembled; (iii) samples were distinguished according to the barcode (exact matching) and primers (≤2 nucleotide mismatch). Finally, we have added the data sets were deposited on the NCBI Sequence Read Archive under accession numbers PRJNA1302308.

5.NMDS and network analyses lack distance metrics, correlation thresholds and FDR correction approach; PCNM/VPA model F and p values are also omitted. Please provide these parameters in Methods and Results.

Response:

(i) We thank the reviewer for this constructive comment. In response, we have added the missing methodological details. Specifically, we clarified that NMDS analyses were based on Bray–Curtis distance (line 161) and reported stress values in the Results (line187).

(ii) For co-occurrence network analysis, in this study, we initially employed Spearman’s correlation with conventional significance testing to construct bacterial co-occurrence networks, which provided an intuitive overview of potential associations among taxa. However, such correlation-based approaches are known to be sensitive to compositionality and multiple testing issues, and thus may overestimate the number of spurious edges. To address this concern, we conducted a new network analysis using a permutation-based null model combined with Benjamini–Hochberg FDR correction. We set the significance threshold at FDR-adjusted p < 0.1 to balance statistical stringency with biological interpretability. In microbial co-occurrence networks, the large number of simultaneous comparisons can make overly conservative cutoffs (FDR < 0.05) too restrictive, potentially excluding ecologically meaningful associations. By adopting a more relaxed yet commonly accepted threshold, we ensured that the retained edges were statistically reliable while preserving sufficient network complexity to capture the underlying ecological interactions.

To assess the statistical significance of inferred associations, we implemented a permutation-based null model with 4,000 randomizations to generate empirical p-values for each potential edge. Multiple testing was controlled by applying the Benjamini–Hochberg procedure, and only associations with false discovery rate (FDR) adjusted p < 0.10 were retained as significant edges. The convergence of results across both correlation-based and null-model-based frameworks supports the robustness of our findings. And we have made corresponding modifications in the methods (section 2.3., line 176-180), conclusions (section 3.2.2., line 255-266), and discussion (section 4.3., line 378-399).

(iii) For PCNM and VPA analyses, we have added F and p values from permutation tests (999 permutations) to the Results section (line 220-222). These details have been incorporated into the Methods (line 170-171).

6.Section 3.2.2 is present but 3.2.1 is missing. Verify section numbering and ensure all figures are cited in order at ≥300 dpi.

Response:

We have carefully re-checked the manuscript and corrected the inconsistencies. Specifically, we revised the section numbering, ensured that all figures are cited sequentially in the text, and updated the figure files to a resolution of ≥300 dpi. These modifications have been incorporated to fully meet the journal’s formatting requirements.

7.Several citations lack DOIs and journal titles are inconsistently abbreviated. Please format all references per PLOS ONE style (full titles + DOI) and remove extra indents.

Response:

In the revised manuscript, we have reformatted all references according to the PLOS ONE style, ensuring the use of full journal titles, consistent formatting, and the inclusion of DOIs where available (see references for details).

8.To help readers visualize the sampling layout and surrounding context, please add a study-area location map (~1:50,000 scale) with coordinates, and cite it in the “2.1 Study area” or Methods section.

Response:

We appreciate the reviewer’s helpful suggestion. In response, we have prepared a study-area location map showing the geographic coordinates of Jiashan County and the distribution of sampling sites. This new figure has been added as Figure 1 in the revised manuscript and is cited in Section 2.1. Experimental design and soil sampling.

To reviewer #2:

Reviewer #2:

1. The manuscript does not adequately position itself within the broader context of literature on agricultural intensification versus ecological restoration. It is therefore necessary to clarify how consolidation differs from conventional practices such as tillage or fertilization.

Response:

We appreciate the reviewer’s valuable comment. In the revised manuscript, we have added corresponding descriptions in the Introduction. Specifically, we now highlight how land consolidation differs from conventional practices such as tillage and fertilization in line 43-51.

2.The experimental design includes sufficient spatial replication; however, the comparison of 36 versus 14 sites is substantially unbalanced. A power analysis (e.g., conducted using G*Power or the pwr package) indicates that to detect a medium effect size (Cohen’s d = 0.5) at a significance level of α = 0.05 with 80% statistical power, a balanced sample size of approximately 26–30 per group is required. Such imbalance in the current design may increase the risk of Type II error and potentially lead to inflated effect size estimates.

Response:

We fully acknowledge the reviewer’s concern that our sampling design was unbalanced (36 LC vs. 14 WL). This limitation was indeed present. The reason for this imbalance lies in the actual field conditions: during our field survey, the distribution of consolidated and non-consolidated paddy plots in the study area was highly uneven, as land consolidation had already been widely implemented locally. Our grouping therefore reflected the real situation of land management rather than an ideal balanced design.

We carefully evaluated whether the imbalance might bias our results. We used PERMANOVA (Bray–Curtis) combined with betadisper to ensure that differences were not simply driven by heterogeneous within-group dispersions. Indeed, the PERMDISP test indicated a significant difference in within-group dispersions (F = 4.19, p = 0.037), suggesting that group variance heterogeneity may have influenced the PERMANOVA results. To address this issue, we conducted balanced subsampling (repeatedly resampling the larger group to match the smaller group across 999 iterations) and re-ran the PERMANOVA as well as complementary analyses, including the neutral community model, niche breadth, and cohesion metrics. The subsampling consistently showed that the main patterns remained robust (Figure S1), and the specific sub sampling results can be found in the attached file (Table S3.csv). The figure shows the results of balanced subsampling, in which the larger group (LC) was randomly resampled to match the sample size of the smaller group (WL). Histograms represent the distributions of key statistics across iterations: (i) PERMANOVA p-value: Distribution of significance levels for community composition differences. (ii) NCM R² difference and NCM m difference: Differences in the fit of the neutral community model (R² and migration parameter m) between groups. (iii) Levins niche width difference: Differences in ecological niche breadth. (iv) Positive and negative cohesion differences: Differences in network connectivity, indicating stronger positive cohesion under LC while negative cohesion differences remain inconsistent. The specific sub sampling results can be found in the attached file (Table S3.csv).

We believe that these additional analyses help to alleviate concerns regarding the potential impact of unbalanced sample sizes on our findings. Nevertheless, we have also addressed this issue in the

---

## [Decision Letter · Decision Letter 1]

30 Sep 2025

Land consolidation drives changes in soil bacterial community structure and promotes positive bacterial interactions

PONE-D-25-26012R1

Dear Dr. Shi,

We’re pleased to inform you that your manuscript has been judged scientifically suitable for publication and will be formally accepted for publication once it meets all outstanding technical requirements.

Kind regards,

Erika Kothe

Academic Editor

PLOS ONE

Additional Editor Comments (optional):

One reviewer indicated that small editorial polishing might be required. This can be done at the page proof stage.

Reviewers' comments:

Reviewer's Responses to Questions

**Comments to the Author**

Reviewer #1: All comments have been addressed

Reviewer #2: All comments have been addressed

2. Is the manuscript technically sound, and do the data support the conclusions?

Reviewer #1: Partly

Reviewer #2: Yes

3. Has the statistical analysis been performed appropriately and rigorously?

Reviewer #1: Yes

Reviewer #2: Yes

4. Have the authors made all data underlying the findings in their manuscript fully available?

Reviewer #1: Yes

Reviewer #2: Yes

5. Is the manuscript presented in an intelligible fashion and written in standard English?

Reviewer #1: Yes

Reviewer #2: Yes

Reviewer #1: The authors have thoroughly revised their manuscript and effectively addressed the concerns raised in previous review rounds. The study is well-structured, methodologically sound, and provides novel insights into how land consolidation alters soil bacterial community structure, assembly mechanisms, and interaction networks in paddy soils. The addition of explicit hypotheses, detailed methodological descriptions, rigorous quality control, and improved data transparency significantly strengthen the work. The discussion acknowledges limitations such as sample imbalance and single time-point sampling while offering clear perspectives for future research. Overall, the manuscript presents valuable and timely findings with both theoretical and practical relevance for microbial ecology and sustainable land management. I recommend acceptance after minor editorial polishing.

Reviewer #2: (No Response)

**Do you want your identity to be public for this peer review?** For information about this choice, including consent withdrawal, please see our Privacy Policy

Reviewer #1: No

Reviewer #2: **Yes: ** Junqiang Zheng

---

## [Editor Report · Acceptance letter]

PONE-D-25-26012R1

PLOS ONE

Dear Dr. Shi,

I'm pleased to inform you that your manuscript has been deemed suitable for publication in PLOS ONE. Congratulations! Your manuscript is now being handed over to our production team.

Kind regards,

on behalf of

Prof. Dr. Erika Kothe

Academic Editor

PLOS ONE